# mTOR Signaling Pathway and Gut Microbiota in Various Disorders: Mechanisms and Potential Drugs in Pharmacotherapy

**DOI:** 10.3390/ijms241411811

**Published:** 2023-07-22

**Authors:** Yuan Gao, Tian Tian

**Affiliations:** College of Life Science and Bioengineering, Beijing Jiaotong University, Beijing 100044, China; 20121607@bjtu.edu.cn

**Keywords:** mTOR, gut microbes, metabolites, therapy

## Abstract

The mammalian or mechanistic target of rapamycin (mTOR) integrates multiple intracellular and extracellular upstream signals involved in the regulation of anabolic and catabolic processes in cells and plays a key regulatory role in cell growth and metabolism. The activation of the mTOR signaling pathway has been reported to be associated with a wide range of human diseases. A growing number of in vivo and in vitro studies have demonstrated that gut microbes and their complex metabolites can regulate host metabolic and immune responses through the mTOR pathway and result in disorders of host physiological functions. In this review, we summarize the regulatory mechanisms of gut microbes and mTOR in different diseases and discuss the crosstalk between gut microbes and their metabolites and mTOR in disorders in the gastrointestinal tract, liver, heart, and other organs. We also discuss the promising application of multiple potential drugs that can adjust the gut microbiota and mTOR signaling pathways. Despite the limited findings between gut microbes and mTOR, elucidating their relationship may provide new clues for the prevention and treatment of various diseases.

## 1. Introduction

mTOR is a conserved serine/threonine kinase with two structurally similar but functionally distinct protein complexes named mTOR complexes 1 (mTORC1) and 2 (mTORC2). mTOR is sensitive to different environmental factors, such as energy and nutrient availability, and is involved in regulating cellular activities such as cell proliferation, cell growth, protein synthesis, and autophagy [1,2,3]. Thus, mTOR plays a central role in regulating cell growth and metabolism in complex physiological processes. In addition, the dysregulation of mTOR signaling has been reported in many diseases such as cancer, diabetes, heart disease, and neurological disorders [2,4,5].

Commensal microorganisms in the human body have gained more and more attention in recent years. The vast majority of commensal microorganisms colonize the gut and differ between individuals due to genetic background, age, sex, diet, and other factors. It has been found that gut microbiota are involved in the regulation of various diseases, such as intestinal diseases, liver diseases, heart diseases, obesity [6], diabetes [7], and Alzheimer’s disease [8], and are associated with physiological processes such as immunity and autophagy. We searched the keywords “mTOR” and “gut microbiota” and obtained 190 results on PubMed and 288 results on Web of Science. These results involved biochemistry, molecular biology, gastroenterology hepatology, pharmacology, pharmacy, and other research fields, as shown in Figure 1. But curiously, only a few of these results demonstrate the direct evidence between mTOR and gut microbiota. And only a few reviews describe the link between the two in disease [9]. Moreover, there are fewer reports on the application of combining different drugs in the treatment of gut microbes or mTOR.

In the present review, we discuss mTOR signaling in detail and review the recent evidence and the underlying mechanisms in which the crosstalk between gut microbes and mTOR are involved in multiple diseases, demonstrating how they function to affect the body’s homeostasis. We also discuss the potential therapeutic drugs that are currently available to regulate gut microbes and the mTOR signaling pathway in different disorders. These will surely benefit patients in the development of new therapeutic strategies in the future.

## 2. mTOR (Mechanistic Target of Rapamycin)

### 2.1. mTORC1 and mTORC2 Structure

mTOR is a member of the phosphoinositide 3-kinase (PI3K)-related protein kinase (PIKK) family, which has serine/threonine protein kinase activity [10,11]. In mammals, mTOR combines with different subunits to form two macromolecular complexes, mTOR complexes 1 (mTORC1) and 2 (mTORC2). Some subunits exist in both complexes, while others are specific components in only one complex (as shown in Figure 2) [5].

mTORC1 is composed of mTOR, mammalian lethal, with SEC13 protein 8 (mLST8) and the regulatory-associated protein of mTOR (RAPTOR) as core components, forming a dimeric structure [12,13,14]. RAPTOR is the defining subunit of mTORC1 and is responsible for recruiting substrates and their subcellular localization [15]. The proteins proline-rich AKT substrate of 40 kDa (PRAS40), DEP domain-containing mTOR-interacting protein (DEPTOR), and the Tti1/Tel2 complex are the remaining components of mTORC1 [1]. In particular, both PRAS40 and DEPTOR are endogenous repressor proteins of mTORC1 [16,17].

The mTORC2-specific combinatorial proteins include rapamycin-insensitive companion of mTOR (RICTOR) [18], a protein observed with RICTOR 1/2 (PROTOR 1/2) [19], mammalian stress-activated protein kinase-interaction protein 1 (mSIN1) [20], and some common subunits of mTOR kinase, such as mLST8, DEPTOR, Tel2, and Tti1 with mTORC1 [2]. RICTOR and mSIN1 are the defining subunits of mTORC2. The N-terminal of mSIN1 is embedded in RICTOR and binds to mLST8 to maintain the stability of mTORC2 [21]. Meanwhile, mSIN1 ensures substrate recruitment [22] and plays an important role in plasma membrane localization and the inhibition of mTORC2 [23,24].

### 2.2. Upstream Regulation of mTORC1

mTORC1 is the more well characterized of the two complexes. It integrates multiple intracellular and extracellular upstream signals, including growth factors, energy inputs, and nutrients. The activation and localization of mTORC1 require the upstream signals of two groups of small G proteins, Ras homolog enriched in the brain (Rheb) and Ras-related GTPases (Rags) [25,26,27]. The Rags form two types of heterodimers, either Rag-A or Rag-B with Rag-C or Rag-D. When the cellular environment contains sufficient cytokines, nutrients, endocrine signals, and energy, mTORC1 is recruited from the cytoplasm to the lysosome via GTP-bound Rag-A or Rag-B (Rag-A/Rag-B^GTP^) and GDP-bound Rag-C or Rag-D (Rag-C/Rag-D^GDP^) and activated by Rheb in the GTP-bound state on the lysosome [28].

mTORC1 is regulated by growth factors and other mitogens, and the tuberous sclerosis complexes (TSC)1/2 play an important role in this process. TSC1/2 acts as a GTPase-activating protein (GAP) on lysosomal Rheb, converting it to an inactive GDP-bound state to inhibit mTORC1 [26,29]. TSC1/2 transmits multiple upstream signals that affect mTORC1. Insulin and insulin-like growth factor 1 (IGF1) phosphorylate TSC2 via the PI3K/AKT pathway, which dissociates TSC from the lysosomal surface and activates mTORC1 [30,31,32]. AKT stimulates the phosphorylation of the endogenous repressor of the mTORC1 complex, PRAS40, to promote Rheb-driven mTORC1 activation in a TSC1/2-non-dependent manner [17,33]. Pro-inflammatory cytokines, such as tumor necrosis factor-α (TNF-α), phosphorylate TSC1 via IκB kinase β (IKKβ), leading to TSC1/2 inhibition, which is similar to the mechanism of growth factors [34]. Classical Wnt signaling activates mTORC1 by inhibiting glycogen synthase kinase 3 (GSK3), and GSK3 phosphorylates TSC2 by initiating phosphorylation in an AMPK-dependent manner [35]. In addition, TSC is inhibited through the regulation of ERK and P90 ribosomal S6 kinase (RSK) [36,37].

mTORC1 is inactive when the environment is low in oxygen and energy, in which case the cellular synthetic response is inhibited. When ATP production is poor, AMP and ADP promote sensitivity to AMP-activated protein kinase (AMPK) activation [38]. AMPK inhibits the mTORC1 complex during energy stress via a twin mechanism: AMPK acts directly on mTORC1 to phosphorylate RAPTOR [39]; it also phosphorylates TSC2 and increases its GAP activity on Rheb, indirectly inhibiting mTORC1 [40]. When oxidative stress is enhanced, thioredoxin 1 (Trx1) in cardiomyocytes interacts directly with mTORC1 to maintain its kinase activity [41]. Hypoxia induces the transcriptional regulation of DNA damage response 1 (Redd1) expression to activate TSC1/2 [42,43]. Notably, DNA damage also induces the expression of various p53 target genes, including the phosphatase and tensin homolog deleted on chromosome 10 (PTEN), TSC1/2, and AMPK, which signal to mTORC1 [44,45]. However, in cardiomyocytes, DNA-damage-inducible transcript 4-like (DDiT4L) inhibits mTORC1 but activates mTORC2 activity [46]. Additionally, cellular hypoglycemic conditions inhibit the binding of hexokinase-II (HK-II) to mTORC1 [47]. Other cellular stresses such as amino acid deficiency, hyperosmolarity, and PH stress induce TSC2 recruitment to the lysosome, thereby negatively regulating mTORC1 [48].

mTORC1 can be activated by amino acids, a process that is dependent on the involvement of Rag GTPases. Mammals have four species, ranging from Rag-A to Rag-D, where Rag-A or Rag-B form obligate heterodimers with Rag-C or Rag-D, respectively. The active conformation of RAG is related to the loading state of the nucleotide when, in the “on” state, RagA/B binds to GTP and RagC/D binds to GDP. At this point, mTORC1 is recruited into the lysosome by RAG and is then activated by Rheb. In the presence of sufficient nutrients, the Rag in the “on” state interacts with RAPTOR to ensure the activation of mTORC1 [49]. When amino acid depletion occurs, Folliculin (FLCN) relocalizes to the lysosome and prevents the exchange of GDP with GTP in Rag-A [50].

mTORC1 perceives amino acid concentrations in lysosomes and envelopes through multiple mechanisms, and these processes require different complexes to deliver amino acid signals to the Rag [51]. Several well-characterized amino acid sensors have been identified. Sestrin2 is a cytoplasmic leucine sensor of the mTORC1 pathway [52]. Under leucine starvation conditions, Sestrin2 binds and inhibits GAP activity toward Rags 2 (GATOR2) [53]; after restoring leucine levels, leucine binds to Sestrin2 and dissociates GATOR2 to activate mTORC1 [54]. The mechanism of cellular arginine sensor for mTORC1 (CASTOR1), a cytoplasmic arginine sensor, is similar to that of Sestrin2. Arginine binds directly to CASTOR1 and relieves the inhibition of GATOR2 [52]. SLC38A9, a lysosomal arginine sensor, transmits state information about amino acids to mTORC1. SLC38A9 interacts with the Rag GTPase–Ragator complex, which is localized to the lysosome, mediating the transport of leucine-based essential amino acids and activating mTORC1 in an arginine-dependent manner [55,56,57]. The S-adenosylmethionine (SAM) sensor, upstream of mTORC1 (SAMTOR), the upstream sensor of SAM, binds to GATOR1 and negatively regulates mTORC1 when methionine is starved or SAM is at low levels [58]. In addition, vesicular H^+^-adenosine triphosphate ATPase (v-ATPase) is required for the amino acid activation of mTORC1 [59].

### 2.3. Substrates and Functions of mTORC1

Once activated by different inputs, mTORC1 responds by acting on different substrates. It not only promotes anabolic synthesis such as protein, nucleic acid, and lipid synthesis; cellular metabolism; and energy expenditure, but also inhibits catabolic processes, including autophagy.

Firstly, mTORC1 phosphorylates eukaryotic initiation factor 4E-binding proteins (4E-BPs) and p70 S6 kinase 1 (S6K1), thereby promoting protein synthesis [60]. The phosphorylation of 4E-BP1 releases eukaryotic translation initiation factor 4E (eIF4E) and promotes the formation of its complex, thereby de-repressing the translation process. Phosphorylated S6K1 phosphorylates its eponymous target ribosomal protein S6 to participate in controlling the transcriptional process during ribosome genesis [61]. In addition, S6K1 augments protein synthesis through the activation of eIF4B and the degradation of programmed cell death 4 (PDCD4) [62,63]. Although both are involved in the control of translation, the role of 4E-BP1 is more prominent [3].

Secondly, mTORC1 controls the synthesis of lipids and nucleic acids required for cell proliferation. To a large extent, mTORC1 acts through the transcription factor sterol regulatory element binding protein 1/2 (SREBP1/2), a substrate that controls the expression of multiple lipid genes. The transcription of SREBP1/2 can be regulated by mTORC1 by modulating the phosphorylation of lipin-1 or in an S6K1-dependent manner. mTORC1 inhibition reduces SREBP1/2 expression, impairing its function and decreasing lipid synthesis [64,65,66]. The phosphorylation of mTORC1 also increases the expression level of peroxisome proliferator-activated receptor γ (PPAR-γ) to promote adipogenesis [67,68]. In addition, mTORC1 promotes lipid biosynthesis by regulating SR protein kinase 2 (SRPK2) to stabilize lipid biosynthetic enzymes [69].

mTORC1 controls the synthesis of new pyrimidines and purines in different cellular models to enrich the nucleotide pool for nucleic acid synthesis, which is essential to maintain DNA replication and RNA synthesis. The de novo synthesis of pyrimidines of mTORC1 is mediated by S6K1, promoting the enzymatic activities of carbamoyl-phosphate synthetase 2 and aspartate transcarbamoylase, dihydroorotase (CAD) [70,71]. mTORC1 stimulates activating transcription factor 4 (ATF4) and mitochondrial tetrahydrofolate cycle enzyme methylenetetrahydrofolate dehydrogenase 2 (MTHFD2) to promote purine synthesis [72].

Moreover, mTORC1 also actively regulates cellular metabolism and ATP production. mTORC1 increases the expression of glycolytic enzymes by activating the transcription and translation of hypoxia-inducible factor 1α (HIF1α), while the activation of SREBPs enhances the pentose phosphate pathway [65]. It has been shown that mitochondrial biogenesis is regulated by mTORC1 by promoting the binding action of transcription factor yin–yang 1 (YY1) with PPARγ coactivator-1a (PGC1-a) [73]. mTORC1 enhances mitochondrial activity through 4E-BP to increase ATP production [74].

In addition to its positive effects on anabolism, mTORC1 negatively regulates catabolism, especially autophagy, to promote growth [75]. Autophagy is the process of the degradation of intracellular components and is considered essential for the organism’s starvation response as well as for organelle renewal. Unc-51-like autophagy-activating kinase 1 (ULK1) and ATG13 are formed in complex with the 200 kDa FAK family-interacting protein (FIP200) and ATG101 to promote autophagy [76]. mTORC1 phosphorylates and inhibits ULK1 and A TG13 to negatively regulate autophagy [77,78,79]. mTORC1 can also bind and phosphorylate UVRAG, thereby enhancing the antagonistic effects of autophagic vesicle maturation in a nutrient-rich environment [80]. When mTORC1 is repressed, the nuclear translocation of transcription factor EB (TFEB) [81] and transcription factor E3 (TFE3) are promoted, facilitating the expression of multiple genes associated with autophagy and lysosomes [82,83,84,85]. Recent studies have shown that mTORC1 directly phosphorylates VAMP8, blocking the formation of the STX17-SNAP29-VAMP8 SNARE complex and inhibiting autophagosome–lysosome fusion [86]. mTORC1 may also affect autophagy by regulating effectors such as death-associated protein 1 (DAP1) [87] and WIPI2 [75].

### 2.4. Upstream Regulation of mTORC2

mTORC2 is activated by growth factors such as insulin through PI3K. The PH domain in mSIN1 can combine with phosphatidylinositol 3,4,5-trisphosphate (PIP3) upon PI3K activation and result in mTORC2 activation [24]. Stimulated by PI3K signaling, the ribosome binds to mTORC2, which is essential for mTORC2 activation [88]. mTORC2 activation also requires the interaction with TSC1/2 independently of Rheb, but the mechanism remains unclear [89].

mTORC2, in addition to being regulated by PI3K/AKT, is also regulated through the mTORC1 negative feedback loop. The degradation of IRS1 induced by mTORC1 and its effector S6K inhibits insulin/PI3K/AKT signaling [90,91,92]. Growth-factor-receptor-bound protein 10 (Grb10), which negatively regulates insulin/IGF-1 signaling, is activated by mTORC1, and mediates the inhibition of PI3K [93,94]. In addition, SK6 promotes RICTOR and mSIN1 activity, thereby inhibiting mTORC2 [95,96]. In contrast, AMP-activated protein kinase (AMPK) directly phosphorylates the complex of mTORC2 and RICTOR and promotes AKT signaling, a process that is independent of mTORC1-mediated negative feedback [97].

Compared with mTORC1, the cellular localization of mTORC2 is more diverse and can localize on the plasma membrane, endoplasmic reticulum, mitochondria, and mitochondria-associated ER membranes (MAM) [98]. However, it remains to be unrevealed how mTORC2 is stimulated at these sites.

### 2.5. Substrates and Functions of mTORC2

mTORC2 regulates a variety of important cellular functions, such as cell structure, metabolism, survival, and proliferation, by regulating members of the AGC family, including AKT, serum, glucocorticoid-induced protein kinase 1 (SGK1), and protein kinase C-α (PKC-α). mTORC2 can phosphorylate its terminal hydrophobic motif Ser473 to directly activate AKT [99]. The deletion of AKT-Ser473 phosphorylation impaired only forkhead box O1/3a (FoxO1/3a) and did not affect TSC2 and GSK3-β [100,101]. mTORC2 activates SGK1 and its substrate N-myc downstream regulated gene 1 (NDRG1) to promote cell survival in hypoxic conditions, which is similar to the effect of FOXO3a [102,103,104]. In addition, mTORC2 mediates cell survival by inhibiting mammalian sterile 20-like kinase (MST1) of the hippo pathway [105,106]. PKC-α is the third AGC kinase activated by mTORC2, which is involved in the regulation of the cytoskeleton and the structure together with RHO GTPases [18,107].

## 3. Gut Microbiota

Trillions of microorganisms have adapted to inhabit the human intestine and form a complex ecological community, which is an indispensable part of the human body [108]. The microbiota obtained at birth develops synchronously with the development of the host and maintains its stability and diversity after adulthood [109]. Because microorganisms are distributed in different regions with the host, diversity is determined by the local environment, including diet [110], high altitude, and other extreme weather [111,112].

The gastrointestinal tract (GI) is composed of the stomach, small intestine (SI), and large intestine (LI). Specific microbiota reside in the unique microenvironment of each section [109]. Five categories of microbiota reside in the stomach of healthy people, namely *Firmicutes*, *Bacteroidetes*, *Actinobacteria*, *Fusobacteria*, and *Proteobacteria* [113]. Gram-positive bacteria and facultative anaerobic bacteria are mainly found in the SI, including *Lactobacilli*, *Enterococci*, and *Streptococci*, and are beneficial to food digestion and nutrition absorption [114]. *Firmicutes* and *Bacteroidetes* mainly settle in the large intestine [115].

In addition to the number, species, and composition of gut microbes, the metabolites of microbes have attracted more attention in recent years. Undigested carbohydrates in the SI are fermented by microorganisms in the LI to produce single-chain fatty acids (SCFAs), namely acetate, propionate, and butyrate [116,117].

Gut microbiota contribute to the defense of intestinal pathogens, nutrition, and energy absorption from the diet, as well as the maintenance of normal immune function [118]. Dysregulation of the normal balance between gut microbiota and the host is involved in many disorders, such as obesity [119], aging [120], inflammatory bowel disease [121], neurological diseases [122], and the occurrence and development of tumors [123]. Recent studies have found that intestinal microorganisms are more closely related to the mTOR signaling pathway (as shown in Figure 3).

### 3.1. Gut Microbiota, mTOR, and Intestinal Diseases

Inflammatory bowel disease (IBD), including ulcerative colitis (UC) and Crohn’s disease (CD), is a chronic, recurrent, intestinal inflammatory disease mediated by immunity. The most common symptoms of these diseases are abdominal pain, diarrhea, and weight loss, and the incidence rate is increasing worldwide [124,125,126]. Intestinal inflammation caused by the Western diet is accompanied by the activation of mTOR, which is contingent on microbial-derived pathogen-associated molecular patterns, in intestinal epithelial cells (IEC), such as endotoxin, poly (I:C), and flagellin [127]. It is worth noting that IL-33 can activate the mTOR pathway of intestinal epithelial cells and promote the induction of REG3γ to facilitate the symbiosis of intestinal microflora [128]. Short-chain fatty acids (SCFAs), a metabolite of intestinal microorganisms, can promote the production of IL-22 by CD4^+^ T cells and IEC. In particular, butyrate can accelerate the phosphorylation of STAT3 and mTOR to upregulate the expression of IL-22 to maintain the stability of the intestinal environment [129]. In addition, extra-intestinal activation of microbiota-specific CD4^+^ T cells and the concomitant inhibition of mTOR metabolism can remove CD4^+^ T memory (TM) cells, thereby preventing colitis [130].

Chronic inflammation is a characteristic of IBD and the driving force of human colon cancer. Studies have shown that the microbial composition of the mouse colon induced by dextran sodium sulfate (DSS) and/or azomethane (AOM) is significantly different from that of the control group. It is worth noting that the abundance of Clostridium and Staphylococcus aureus is increased. Meanwhile, PI3K/AKT/mTOR, fatty acid metabolism, and oxidative phosphorylation signals were found to be upregulated in DSS/AOM mice via gene expression profiling [131]. *Clostridium butyricum* promotes the upregulation of AKT/mTOR and downstream molecule p70 ribosomal protein S6 kinase expression and alters the production of anti-inflammatory cytokines to exert a protective effect on intestinal barrier function [132]. In contrast, *Lactobacillus X12* inhibited the growth of colorectal cancer cells by inhibiting mTOR and regulating cell-cycle-associated proteins p27 and E_1_ [133]. In the genesis and development of colon cancer, the imbalance of gut microbiota is accompanied by the release of endotoxin lipopolysaccharide (LPS) [134]. With the increase in the LPS level, Cathepsin K (CTSK), a metastasis-related secretory protein secreted by tumors, can interact with macrophage membrane receptor Toll-like receptor 4 (TLR4) to activate the mTOR pathway. The TLR4-mTOR-dependent pathway accelerates the M2 polarization of tumor-associated macrophages to promote the progression of colorectal cancer [135].

### 3.2. Gut Microbiota, mTOR, and Liver Diseases

The gut microbiota and mTOR-signaling pathways have been shown to play an important role in liver diseases [136,137,138]. The inoculation of mice treated with a high-fat diet with feces from patients with non-alcoholic fatty liver disease (NAFLD) results in aggravated liver injury [139]. Recently, a study has shown that NF73-1, an *E. coli* isolated from the intestines of NASH patients, can enter the liver through TLR2/NLRP3, induce M1 macrophages, and finally promote the development of NAFLD, with the activation of mTOR-S6K1-SREBP-1/PPAR-α [140]. In fact, the abundance of microbiota is low and its diversity is less in mice with NAFLD. Interestingly, Alisol B 23 acetate (AB23A) could rebalance the gut microbiota, especially reducing the abundance of *Firmicutes*/*Bacteroidaeota* and *Actinobacteriota*/*Bacteroidaeota*. During this process, AB23A plays the role of a probiotic on NAFLD by inhibiting the activity of mTOR, TLR4, and NF-κB [141]. Similarly, *Lactobacillus rhamnosus*, which is considered to be one of the most extensive probiotics, has also been revealed to play an ameliorative role in alcoholic fatty liver. The combination therapy of *Lactobacillus rhamnosus* culture supernatant and bone marrow mesenchymal stem cells (BMMSCs) can accelerate autophagy and improve alcoholic fatty liver disease by inhibiting the PI3K/mTOR pathway [142]. In hepatocellular carcinoma cell line Huh-7, Pant et al. found that butyrate, a short-chain fatty acid produced by gut microbiota during anaerobic fermentation, inhibited the phosphorylation of AKT and mTOR by inducing reactive oxygen species (ROS), thereby inducing the autophagy of liver cancer cells [143]. In general, the dysregulation of the gut microbiota and mTOR signal will promote the progression of liver diseases, in which autophagy and immune cells are also involved.

### 3.3. Gut Microbiota, mTOR, and Heart Disease

The gut microbes and mTOR signaling pathways are both involved in heart disease. Butyrate, a metabolite of gut microbiota, significantly inhibits the PI3K/AKT/mTOR pathway while enhancing ATG5-mediated autophagy in the murine STC-1 enteroendocrine cell line [144]. When mTOR is activated by PI3K and AKT, it can inhibit autophagy by regulating the activity of ULK1 [145]. Urolithin B (UB), a metabolite of gut microbiota, can inhibit autophagy through the AKT/mTOR/ULK1 pathway to play a protective role in myocardial ischemia–reperfusion injury in rats. More specifically, the p62/Keap1/Nrf2 signaling pathway protects against oxidative stress and caspase 3-dependent apoptosis [146]. Similarly, this protective effect was also confirmed in the mouse model of myocardial infarction (MI). UB inhibits cardiomyocyte apoptosis by activating AKT/mTOR pathway and simultaneously suppresses NF-κB to reduce the occurrence of arrhythmia after hypoxia [147].

### 3.4. Gut Microbiota, mTOR, and Other Diseases

The interaction between gut microbiota and mTOR is also reported in other diseases in which autophagy, energy metabolism, and immunity are involved.

AMP-activated protein kinase (AMPK), a key energy receptor, can inhibit the formation of auto phagosomes caused by mTOR [79]. Sodium butyrate can activate the phosphorylation of AMPK in the renal tissue of rats with diabetic mellitus (DM) to inhibit mTOR and increase the number of autophagosomes, thus aggravating the kidney injury of rats with DM [148]. In the rat model of diet-induced obesity (DIO), Xiexin Tang (XXT) promoted the production of SCFAs by intestinal flora and the expression of AMPK to inhibit the activation of the mTOR signaling pathway, to adjust the disorder of lipid metabolism, and to reduce the systemic inflammatory response [149]. In addition, both IL-37 and alanylglutamine could increase the expression of AMPK and reduce the expression of mTOR to alleviate chronic inflammation in mouse models of allergic asthma and atopic dermatitis (AD), respectively. In this process, the diversity and metabolites of intestinal microorganisms are regulated. The difference is that IL-37 increases autophagy-related IC3B and decreases autophagy-related ubiquitinated protein p62, and the NF-κB and STAT3 signaling pathways may be involved in the treatment of alanylglutamine [150,151]. A taxonomical evaluation of the gut microbiota revealed an elevated abundance of *phyla Proteobacteria* and *Firmicutes* in patients with neurodegenerative diseases [152]. The ketogenic diet downregulates mTOR protein expression to delay cognitive deterioration, but the direct link between gut microbial changes and mTOR is not clearly stated [153]. A recent study showed that environmental low-dose radiation (LDR) compromises the intestinal barrier and increases PA in the organism. This is accompanied by the increased expression of PYCR1 in the liver, the inhibition of the IRS-1/AKT/mTOR axis, and an accompanying increase in Parabacteria in the gut microbiota, leading to the impairment of HFD-induced obesity and insulin resistance [154]. In mouse models of high-fat diets, the simultaneous inhibition of mTORC1 and mTORC2 could aggravate intestinal inflammation and destroy blood glucose homeostasis, while the specific inhibition of mTORC1 could alleviate intestinal inflammation and improve glucose tolerance. Interestingly, the chronic inhibition of mTORC2 contributed to the changes in gut microbiota caused by high fat, such as *Turicibacter* and unclassified *Marinilabiliaceae* [155]. The long-term inhibition of mTOR can prolong the lifespan of mice and mildly change intestinal metagenes, which are related to immune cells [156]. Moreover, *Prevotella Copri* obtained from pig intestines can promote a chronic inflammatory response and fat deposition through TLR4 and mTOR signaling pathways after its inoculation in mice [157].

### 3.5. Gut Microbes and mTOR in the Analysis of Big Data

The human intestinal microbes are so numerous and diverse that they have been called the “second genome”. A microbial census Begins with 16s rRNA gene sequencing, and PCR is applied to amplify 16s rRNA; then, sanger sequencing is utilized two or three times to complete the gene sequencing. In order to make the sequencing more accurate, second-generation sequencing and third-generation technologies were developed to make the study of microbiota genomics more efficient and rapid [158].

In order to understand the impact of gut microbes on human health, researchers analyzed 124 human-derived feces through macro genome sequencing to create a genomic catalog of the human gut microbiome, covering most of the prevalent gut microbial genes in humans [159]. More studies are delving into the role of microbiota in different health and disease conditions through transcriptomics and metabolomics. An analysis of microorganisms in feces using 16S rRNA sequencing revealed a relative decrease in bacterial abundance in mice with colorectal cancer [160]. In addition, an analysis of colonic tissues using whole-transcriptome profiling revealed that enhanced PI3K-AKT-mTOR signaling in colorectal cancer mice was accompanied by a significant increase in the expression level of phosphorylated S6 ribosomal protein (a downstream target of the mTOR pathway) [131]. In 129 stool samples from NAFLD patients consuming a high-carbohydrate diet, alterations in intestinal microbial species were found, with a significant increase in the abundance of *Enterobacteriaceae* and a significant decrease in the abundance of *Ruminococcaceae* compared to 75 normal samples. Meanwhile, an analysis of 90 liver transcripts revealed that the expression of SREBF2 and mTOR increased with the enhancement of NAFLD activity [161].

## 4. Application of Multiple Drugs Affecting Gut Microbes and mTOR in the Treatment of Different Diseases

In recent years, further studies on the direct or indirect interaction between gut microbiota and mTOR in various diseases have been published. The significance of a variety of physiological activities in the body, including metabolic response, autophagy, and immune response, is also shown above. Many kinds of drugs have demonstrated promising evidence in preclinical studies in different diseases, as shown in Table 1. More effective treatment methods need further evaluation of their effectiveness and safety in clinical trials.

### 4.1. Treatment of Intestinal Diseases

Some drugs have already been proven to have a potential role in animal models of intestinal diseases such as IBD and CRC. Ming-hui Jin et al. used polystyrene nanoplastics (PS) to stimulate enterotoxicity in mice, resulting in intestinal microbial disruption and colonic tissue damage. Maltol treatment promoted AMPK phosphorylation activity and inhibited mTOR phosphorylation in the colon after PS exposure, promoting TFEB entry into the nucleus to mitigate autophagy-dependent apoptosis. In addition, oral maltol decreased the relative abundance of *Bacteroidetes* but increased the relative abundance of Firmicutes and restored the number of known SCFA-producing bacteria, thereby restoring gut microbial composition [162]. Mu Xia Li et al. revealed the mechanism of Huangqin decoction (HQD) in the treatment of gastrointestinal diseases such as UC. As a traditional Chinese medicine therapy, HQD could improve the clinical performance of the DSS-induced UC model, inhibit the inflammatory response in vivo, and rebalance the gut microbiota. HQD treatment activated PI3K/AKT/mTOR signaling by upregulating amino acid metabolism and improved the barrier function of the intestinal epithelial [163,164]. In contrast, Rhein inhibited pro-inflammatory factors by inhibiting the PI3K/AKT/mTOR pathway, thereby alleviating enteritis. In the process, Lingling Dong et al. found that Rhein treatment resulted in the downregulation of *Enterobacteriaceae* and *Turicibacter* and the upregulation of *Unspecified-S24-7* and *Rikenellaceae*, which were correlated with pro-inflammatory factors [165]. Dandan Wang et al. found that a polysaccharide isolated from Panax ginseng (GP) reduced the intestinal injury of DSS-induced colitis in rats. GP treatment increased the diversity of the microbial community, improved the compositions of gut microbiota, reduced the phosphorylation level of mTOR, and activated autophagy to inhibit inflammation [164]. In addition, SCFAs and metformin (MTF) can regulate intestinal immunity to prevent colitis and have potential therapeutic applications [129,130].

The PI3K/AKT/MTOR signaling pathway plays a key role in a variety of cancers, including CRC, such as cell proliferation, cell metastasis, and cell survival [166,167]. Theabrownin (TB) inhibited the development of CRC by decreasing *Bacteroidceae* and *Bacteroides* associated with CRC and increasing the production of SCFAs, thereby inhibiting cell proliferation through the suppression of PI3K/AKT/mTOR phosphorylation [168].

### 4.2. Treatment of Liver Diseases

In the HFD-induced rat model of NAFLD, fecal levels of *Firmicutes*, *Bacteroidetes,* and short-chain fatty acids returned to normal with treatment with *L. reuteri* + MTZ alone or in combination with MTF. More precisely, combined therapy prevented steatosis and the progression of liver injury by inducing autophagy via p-AKT/mTOR/LC-3Ⅱ pathways in the liver [169]. Fan Xia et al. found that AB23A not only reduced the abundance of *Firmicutes*/*Bacteroidaeota* and *Actinobacteriota*/*Bacteroidaeota* but also decreased the activities of mTOR and TLR4 to prevent the progress of NAFLD [141]. Furthermore, the combined LGG-s and BMMSC treatment also inhibited the PI3K/mTOR signal to accelerate autophagy, which has the potential to alleviate alcoholic steatohepatitis [142]. Interestingly, in the HFD-induced metabolic syndrome, Zhenzhen Deng et al. found that low-molecular-weight fucoidan fraction LF2 and MTF have similar effects on gut microbiota, increasing the proportion of *Verrucomicrobia* and enriching the abundance of *Akkermansia muciniphila*. LF2 promoted the phosphorylation of PI3K and AKT in a dose-dependent manner but reversed the over-activation of mTOR, thereby improving lipid metabolism [170].

According to the report, the gut microbiota can regulate the immune response of hepatocellular carcinoma (HCC); thus, readjusting the gut microbiota could be a potential option for HCC treatment [171]. Butyrate, considered as a potential candidate drug for the treatment of liver cancer, could inhibit the phosphorylation of AKT and mTOR through reactive oxygen species, resulting in the upregulation of autophagy proteins beclin 1, ATG 5, and LC3-Ⅱ, thereby promoting the formation of autophagy bodies [143]. Curcumin can significantly sensitize hepatoma cells to 5-FU cytotoxicity and increase the apoptosis rate through synergistic effects. The gut microbiota facilitates the oral utilization of curcumin in vivo and enhances the chemo-sensitivity of hepatocellular carcinoma cells to 5-FU by blocking the PI3K/AKT/mTOR signaling pathway in vitro [172].

### 4.3. Treatment of Other Diseases

Probiotics regulate the PI3K/AKT/mTOR signaling pathway, which is beneficial for coordinating the immune response. Probiotics fermentation technology (PFT) activated the PI3K/AKT signal transduction pathway but inhibited the glycogen synthase kinase-3β (GSK-3β) and mTOR signal; its potential role in the treatment of Alzheimer’s disease was parallel to that of pioglitazone [173]. Additionally, there are reports that *Aronia melanocarpa* polysaccharide (AMP) activates PI3K/AKT/mTOR signaling pathway and its downstream apoptotic protein family, inhibits brain-cell apoptosis, and enriches intestinal beneficial bacteria to delay aging, which had a similar function to MogrosideV and its metabolite 11-oxo-mogrol [174,175]. In contrast to the mechanism of action of AMP, Xiexin Tang (XXT) ameliorates obesity by promoting the activity of key enzymes for the synthesis of SCFAs and inhibiting AMPK while activating mTOR signaling [149].

Ophiopogonin D (OPD) can increase the abundance of *Bacteroidetes*, reduce the relative abundance of *Firmiuts*, inhibit the phosphorylation of mTOR and the expression of SREBP1 and SCD1 to alleviate fat metabolism, and result in the prevention of atherosclerosis and metabolic syndrome [176]. The mechanism in which β-hydroxyβ-methylbutyrate (HMB) functions through the Bacteroidetes–acetic acid–AMPKα axis to reduce the lipid metabolism of Bama Xiang mini-pigs was somewhat similar [177]. In addition, *Flammulina velutipes* polysaccharide (FVP) affected the abundance of gut microbiota, especially the Bacteroidetes phylum and the Muribaculaceae family, and upregulated the mTOR signaling pathway in cardiac tissue [178]. However, the specific mechanism remains to be determined.

The oral administration of the bruceae Frutus oil (BO), under the influence of gut microbiota, inhibited breast cancer. At the same time, BO changed the dominant strains of gut microbiota and promoted mTOR activity, leading to the inhibition of autophagy [179]. In contrast, 20 (s)-ginsenoside Rh2 (grh2) played an anti-tumor role by inhibiting PI3K/AKT/mTOR signal [180]. Both an engineered resistant starch (ERS) diet and a ketogenic diet (KD) reduced mTOR phosphorylation and regulated microorganisms [181,182]. Diet may serve as a synergistic approach to improve the treatment of diseases.

The direct interaction between mTOR and intestinal microorganisms provides potential ideas for treatment. Firstly, microencapsulated rapamycin (eRapa), the best pharmacological mTOR inhibitor studied in the study of lifespan and health extension, had strong immune effects and could gently change intestinal metagenes, which is worthy of further study [156]. Then, resveratrol, a specific inhibitor of mTOR complex 1, alleviated the changes in intestinal microflora in diet-induced obese mice [155]. Furthermore, the microflora metabolite SCFAs activated mTOR and STAT3 of IEC to produce antimicrobial peptides to balance the intestinal environment [183].

Overall, in view of the limitations of current treatment, more drugs can only be used as a potential choice for disease treatment, which has broad clinical application prospects in the future.

**Table 1 ijms-24-11811-t001:** List of drugs affecting the microbiota and the mTOR signaling pathway in different disorders.

Name	Disease	Pathway Affected	Changes of Gut Microbiota	Cell Response
Maltol [162]	——	↑AMPK ↓mTOR	↑*Firmicutes*, ↓*Bacteroidetes*	↓Apoptosis
HQD [163]	UC	↑PI3K/AKT/mTOR	↑*Firmicutes*, *Bacteroidetes*	↑amino acid metabolism, p-S6 and p-4EBP1 ↓Apoptosis
Rhein [165]	UC	↓PI3K/AKT/mTOR	↑*Unspecified-S24-7, Rikenellaceae* ↓*Enterobacteriaceae, Turicibacter*	↓pro-inflammatory cytokines
P. ginseng [164]	IBD	↓mTOR, TLR4, NF-kB	↓Gram-negative bacteria	↑Autophagy ↓p62
SCFAs [129]	CD and UC	↑mTOR, STAT3	against enteric infection of *Citrobacter rodentium*	↑HIF1α, AhR, IL-22 ↓Gpr41, HDAC
TB [168]	CRC	↓PI3K/AKT/mTOR	↓Bacteroidceae and Bacteroides ↑Prevotellaceae and Alloprevotella	↑cyclin D1 protein, cleaved caspase 3
SCFAs [183]	——	↑mTOR, STAT3	——	↑AMP, RegIIIγ, β-defensins
L. reuteri + MTZ [169]	NAFLD	↓mTOR, AKT	↑*Akkermansia muciniphila*, *Firmicutes*, butyrate	↑Autophagy, LC-3II ↓LPS, NF-kB, TNF-α
AB23A [141]	NAFLD	↓mTOR, TLR4, NF-kB	↓*Firmicutes*/*Bacteroidaeota*, *Actinobacteriota*/*Bacteroidaeota*	↑ZO-1, occludin
LGG-s and BMMSC [142]	Alcoholic liver disease	↓PI3K/mTOR, PI3K/NF-kB	——	↑Autophagy ↓NKB cells, TFH cells
LF2 [170]	METS	↓PI3K/AKT/mTOR	↑*Verrucomicrobia*, *Akkermansia muciniphila*	↓SREBP-1c, PPARγ
Butyrate [143]	HCC	↓mTOR, AKT	——	↑ROS, Autophagy: beclin 1, ATG 5, LC3-II
Curcumin [172]	HCC	↓PI3K/AKT/mTOR	↑family Helicobacteraceaeorder, order Campylobacterales, and genus Helicobacter and Campylobacteria	↑apoptosis
PFT [173]	AD	↑PI3K/AKT ↓mTOR, GSK-3β	——	↓oxidative stress, inflammation
AMP [174]	Brain aging	↑PI3K/AKT/mTOR ↓AMPK/SIRT1/NF-κB	↑*Bacteroides* ↓*Firmicutes*	↓apoptosis, NLRP3
MogV and 11-oxo-mogrol [175]	neuronal damages	↑AKT/mTOR	——	↑neurite outgrowth ↓apoptosis, [Ca2^+^]_i_ release
OPD [176]	atherosclerosis	↓mTOR/SREBP1/SCD1	↑*Bacteroidetes*, *Faecalibaculum* ↓*Firmicutes*, *Ileibacterium*	↑insulin resistance ↓lipid metabolism
HMB [177]	Obesity	↑AMPKα, Sirt1, and FoxO1 ↓mTOR	↑*Bacteroidetes*, acetic acid	↓lipid metabolism
XXT [149]	Obesity	↑AMPK ↓mTOR	↑key synthetic enzymes of SCFAs	↑energy expenditure:PGC-1α, UCP-2 ↓energy intake
FVP [178]	Heart	↑mTOR, etc ↓AMPK, PI3K-Akt, etc	↑*Bacteroidetes*, *Muribaculaceae*	↑Immunity
BO [179]	TNBC	↑mTOR	↑*Candidatus Melainabacteria* bacterium *MEL.A1*, *Ndongobacter massiliensis*, *Prevotella ruminicola*	↓Autophagy Regulate amino acid metabolism
GRh2 [180]	T-ALL	↓PI3K/AKT/mTOR	↑*Bacteroidetes*, *Verrucomicrobia* ↓*Firmicutes*, *Proteobacteria*	↑Immunity, tight junction proteins, antimicrobial peptides, IgA
ERS Diet [181]	PC	↓mTOR, ERK1/2	↑diversity of microbiota ↑Formate, Lactate ↓Propionate	↓Proliferation
Resveratrol [155]	Obesity and Diabetes	↓mTOR	↓*Lactococcus*, *Clostridium* XI, *Oscillibacter*, and *Hydrogenoanaerobacterium*	↑insulin resistance
ERapa [156]	Longevity	↓mTOR	Alteration of gut metagenomes	Regulate T, B, myeloid, and innate lymphoid cells

Arrows indicate upregulation or downregulation.

## 5. Conclusions and Future Perspective

The mTOR signaling pathway plays a significant role in various physiological processes in the cell. The gut microbiota and its complex metabolites regulate a wide range of host functions, including autophagy, fatty acid, oxidative phosphorylation, and immune response [184,185,186]. The dysregulation of the gut microbiota is associated with various disorders, such as cancers, diabetes, and inflammatory diseases, in which the mTOR signaling pathway is also involved. In the present review, we introduced the mTOR signaling pathway, the correlation between gut microbes and mTOR, their function and mechanisms in multiple diseases (the crosstalk between the two is shown in Figure 3), and the possible treatments utilizing microbiota and mTOR inhibitors in these diseases.

Currently, there are still unsolved problems that remain to be elucidated regarding the study of the gut microbiota and mTOR. Most existing studies suggest a phenomenal association between gut microbes and mTOR, with disease development causing changes in composition in the gut microbiota or the regulation of its metabolites, accompanied by the inhibition or activation of mTOR [9,187], as demonstrated in several reports [150,162,165,168,170]. Only a few show the direct correlation between mTOR and the modulation of gut microbiota [155]. We are still far from understanding the underlying mechanism between mTOR and gut microbiota.

The translation of basic mTOR and microbiota studies to the clinical setting remains challenging. Most studies focus on the evidence in animal models, and little is known about mTOR in clinical trials. More data need to be verified in clinical conditions. As the inhibition of mTOR does not always lead to protective effects, it may also be controversial. Furthermore, due to the complexity of the organism, mTOR activity is increased in some pathological conditions and decreased in others, making it difficult to estimate the exact effects of therapeutic intervention. More in-depth systematic studies are greatly needed to elucidate how mTOR is associated with gut microbes, to produce more conclusive and valid results, and to further explore new drug candidates for the treatment of relevant diseases. Targeting therapies toward mTOR signaling combined with probiotic administration to adjust gut microbiota will result in better efficacy in clinical trials in the future.

## Figures and Tables

**Figure 1 ijms-24-11811-f001:**
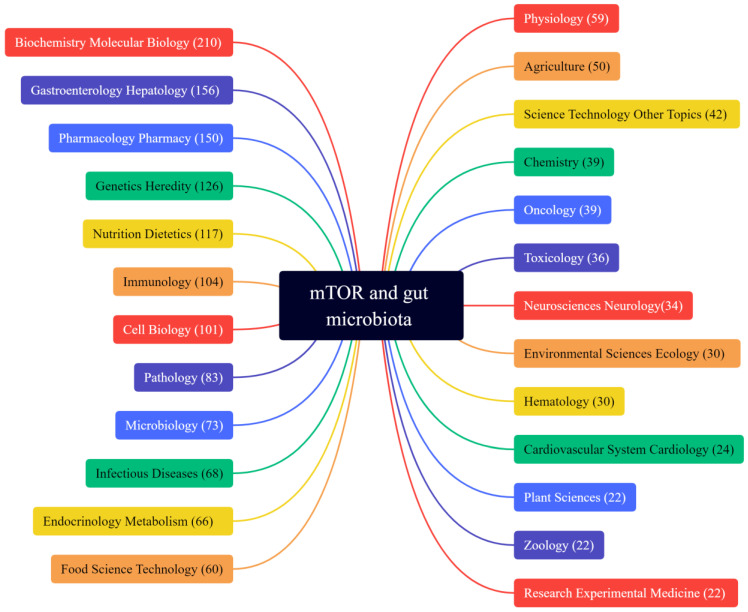
The analysis of Web of Science search results for the keywords “mTOR” and “gut microbiota”. The web search for mTOR and gut microbiota yielded 288 results, which were analyzed and found to cover 25 research directions, including biochemistry, molecular biology, gastroenterology, hepatology, pharmacology, pharmacy, genetics, heredity, and others.

**Figure 2 ijms-24-11811-f002:**
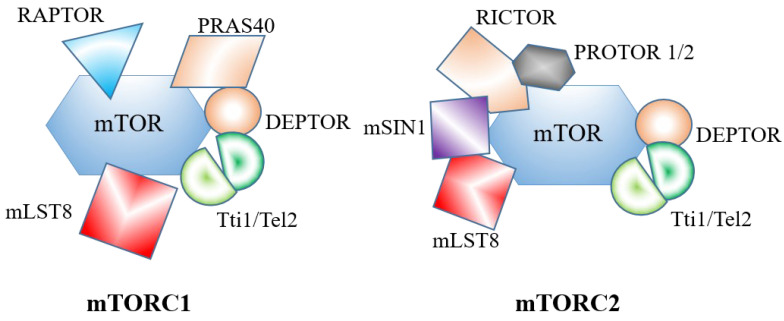
Schematic diagram of mTORC1 and mTORC2 structures. TORC1 and mTORC2 share components mTOR, mLST8, DEPTOR, and Tti1/Tel2 complex. mTORC1-specific subunits include PRAS40 and RAPTOR. mTORC2-specific proteins include RICTOR, PROTOR 1/2, and mSIN1.

**Figure 3 ijms-24-11811-f003:**
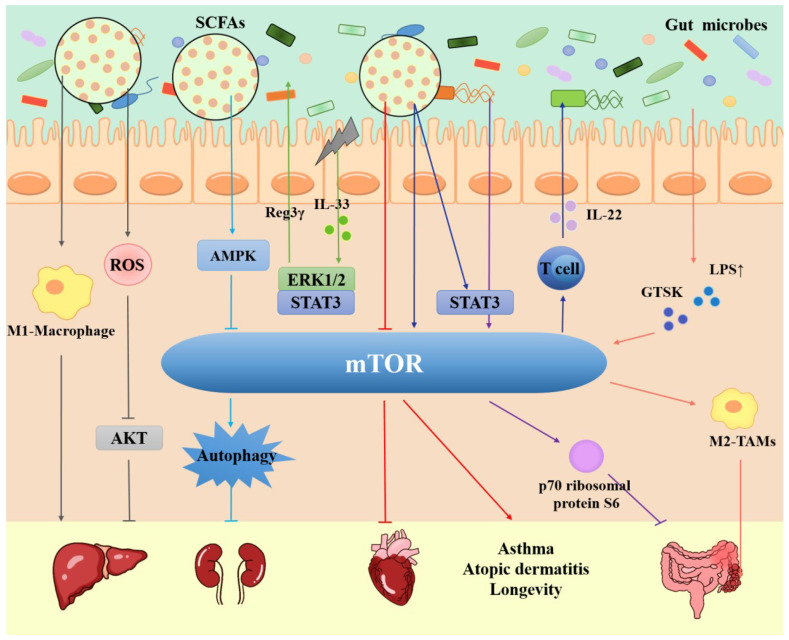
Schematic representation of crosstalk between gut microbes and the mTOR signaling pathway in different diseases. Gut microbes directly or indirectly stimulate mTOR through changes in their own abundance or metabolites (mainly SCFAs), integrating intra- and extracellular signals, which in turn influence disease development. A detailed description of these interactions is provided in the main text. Arrows represent activation, and horizontal lines indicate inhibition. The differently colored lines represent different regulatory mechanisms. Abbreviations: AMPK: AMP-activated protein kinase; ERK1/2: Extracellular-signal-regulated kinases 1/2; ROS: Reactive oxygen species; SCFAs: Short-chain fatty acids; IL33: Interleukin-33; IL-22: Interleukin-22; CTSK: Cathepsin K; LPS: lipopolysaccharide; TAMs: Tumor-associated macrophages.

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
