# Peer review of "mTOR Signaling Pathway and Gut Microbiota in Various Disorders: Mechanisms and Potential Drugs in Pharmacotherapy"

_ijms, 2023, doi:10.3390/ijms241411811_

Round 1

Reviewer 1 Report

The authors made a Review on the mTOR and gut microbiota in various disorders:mechanisms and potential drugs. Please see my comments bellow:

Aim of the study is missing. It is clear what you have done in this study, but why for? Novelty is missing.  The aim of the paper must be presented separately in the last paragraph of the Introduction, after L41, and needs to be addressed from the perspective of describing the contribution to the field under analysis and the elements of scientific novelty presented.

Before inserting references, a free space is needed. Revise the entire manuscript in this regard.

Based on which criteria have you chosen the references? Some of them are very old.

Which is the impact of the topic in the literature? I suggest that the authors will make a short search (graphically) related to the impact of the topic on the general literature (using i.e. Web of Science). Providing a scientometric/bibliometric figure would be interesting in justifying the “novelty”/impact of the topic and will enrich the graphical part of your research. Which keywords and Boolean operators have been used in this search, must be explained. “Treemap” type figure should be presented.

In chapter 3 please describe the interaction between metformin administration, gut microbiota administration and the activity of mTor complex. I suggest checking and referring to DOI: 10.1016/j.lfs.2021.119311

In chapter 4. Please discuss the implications of mTor complex in colorectal cancer and how modifications of gut microbiota by diet quality of probiotic administration can reduce the risk of colorectal cancer. You can check PMID: 26662146 and PMID: 33802777

1 figure and 1 table are far too few for describing the topic.

Check the Instructions for authors https://www.mdpi.com/journal/ijms/instructions regarding abbreviations: Acronyms/Abbreviations/Initialisms should be defined the first time they appear in each of three sections: the abstract; the main text; the first figure or table. When defined for the first time, the acronym/abbreviation/initialism should be added in parentheses after the written-out form. The Abbreviations section must be removed.

Good English.

Reviewer 2 Report

Review: mTOR and gut microbiota in various disorders:mechanisms and 2 potential drugs by Yuan Gao and Tian Tian describes The mammalian or mechanistic target of rapamycin (mTOR) in a context of peripheral organs and related diseases, considering the mechanisms involved in the common mTOR pathway and the microbiota. The manuscript is interesting and addresses an important aspect of the usefulness of the mTOR pathway in the treatment of the diseases described. I found several errors and omissions and ask authors to correct them:

1. Introduction should be expanded a bit with a more detailed description of the cited literature

2. Paragraph 2. Mechanistic target of rapamycin (mTOR), 2.1. mTORC1 and mTORC2 structure should be enriched with a figure showing the construction of mTOR

3. Paragraph 2.2. Upstream regulation of mTORC1, line 75: please elaborate on the description based on the ref. [22]

4. The description of the abbreviation "GATOR" is missing from both the text and the list of abbreviations at the end of the publication. Please check the correctness of refs [51] and [52] in the passage between lines 126-136

5. line 174: ‘…production to.” ? a typo?

6. Line 182: lack of references

7. Line 254: lack of references

8. At the end of the paragraph 3.4. Gut microbiota, mTOR and Other Diseases, please include a brief message to readers that the topic has not been exhausted, as there is a huge amount of research data on mTOR/microbiota/central nervous system diseases involving glutamate as a excitatory neurotransmitter, but the topic is not covered in this article.

9. in the title of Table 1 there was a typo - no spaces between words

10. Line 531: lack of references (a few)

11. please carefully check that all refs are relevant to the text being described

Round 2

Reviewer 1 Report

No novelty of the paper.

Minor English revision.
